# Development and Validation of the Midwifery Interventions Classification for a Salutogenic Approach to Maternity Care: A Delphi Study

**DOI:** 10.3390/healthcare12222228

**Published:** 2024-11-08

**Authors:** Giulia Maga, Arianna Magon, Rosario Caruso, Lia Brigante, Marina Alice Sylvia Daniele, Silvia Belloni, Cristina Arrigoni

**Affiliations:** 1Department of Biomedicine and Prevention, University of Rome Tor Vergata, 00133 Rome, Italy; giuliamaga01@gmail.com; 2Health Professions Research and Development Unit, IRCCS Policlinico San Donato, 20097 San Donato Milanese, Milan, Italy; 3Department of Biomedical Sciences for Health, University of Milan, 20133 Milan, Italy; 4Department of Women’s and Children’s Health, Faculty of Life Sciences and Medicine, King’s College London, London WC2R 2LS, UK; lia.brigante@rcm.org.uk; 5Department of Midwifery and Radiography, School of Health and Psychological Sciences, University of London, London EC1V 0HB, UK; marina.daniele@city.ac.uk; 6Department of Public Health, Experimental and Forensic Medicine, Section of Hygiene, University of Pavia, 27100 Pavia, Italy; silvia.belloni@unipv.it (S.B.); cristina.arrigoni@unipv.it (C.A.)

**Keywords:** midwifery, women and newborns health, maternity care, quality indicators, Delphi

## Abstract

**Background/Objectives:** This study aims to develop and validate a Midwifery Interventions Classification (MIC), which is an evidence-based, standardized taxonomy and classification of core midwifery interventions based on a salutogenic perspective for maternity care. **Methods:** This study described the consensus process up to the results regarding the validation of the MIC through a two-round Delphi survey involving three panels of stakeholders: Midwives, Healthcare Researchers, and Maternity Service Users. **Results:** The resulting MIC comprises 135 core midwifery interventions classified into Direct Midwifery care (n = 80 interventions), Indirect Midwifery Care (n = 43 interventions), and Community Midwifery Care (n = 12 interventions), reaching an overall consensus rate among experts equal to 87%. These interventions were, therefore, adapted specifically for the Italian midwifery care context, with potential for international transferability, implementation, and scalability. **Conclusions:** The MIC is pivotal to boosting quality improvement, education, and comparable data collection for research, sustaining midwives’ role in promoting optimal health for women, newborns, and families at large.

## 1. Introduction

The maternity care received by women, newborns, and families is a core issue of public health internationally, which also includes how it is organised, who it is delivered by, and its quality and content [1,2,3,4]. In the maternity care setting, the midwifery continuity care model is supported by evidence as the gold standard approach [1,5]. Thus, midwife continuity of care models provide care from the same midwife or team of midwives during pregnancy, birth, and the early parenting period in collaboration with obstetrics and specialist teams when required [1]. Midwives have specific contributions to make about skilled supportive and preventive care for all, the promotion of physiological reproductive processes, first-line management of complications, and skilled emergency care. These aspects are intended in the context of respectful care tailored to needs and work to strengthen women’s capabilities, and they are integrated across facility and community settings [6]. Sandall et al. [1] established that women and babies who received midwife continuity of care models compared to those receiving other models of care are more likely to achieve optimal health outcomes, such as spontaneous vaginal birth and a positive experience, and are less likely to experience unnecessary interventions, such as episiotomy [1,7].

Reducing childbirth medicalisation, which reflects on unnecessary and frequently overused interventions [6,8], is the goal of current midwifery practice, which focuses on promoting a salutogenic approach to maternity care [8,9,10,11]. *The Lancet*’s series on Midwifery highlighted that midwives are pivotal in supporting a holistic and woman-centered care approach, focusing on factors supporting human health and well-being rather than factors causing disease, and promoting childbirth as a normal life event [6]. However, most maternity care quality indicators measured by healthcare systems primarily focus on disease prevention or adverse events (e.g., morbidity, mortality, and disability), with less emphasis on collecting data aimed at promoting health and wellness-focused maternity care practices [9,11]. Thus, it becomes crucial to outline quality indicators for midwifery interventions and related outcomes within a salutogenic approach to maternity care, such as improving exclusive breastfeeding rates, enhancing maternal satisfaction and well-being with care, and supporting women and families in their transition to parenthood [12,13].

A growing number of studies in the literature currently focus on describing the aspects and qualities of salutogenic midwifery care. For instance, the study by Mathis et al. has contributed to synthesizing a preliminary framework of the salutogenic aspects of midwifery care, particularly concerning the cognitive, behavioral, and emotional components of health promotion during childbearing [14]. Furthermore, the studies by Smith et al. and Maga et al. have contributed to defining a set of salutogenically oriented outcomes during the antenatal, birth, and postnatal periods [10]. Thus, exploring these salutogenic aspects encourages the definition of midwives’ practices and their role in maintaining health and well-being in the maternity care population. Nonetheless, the salutogenic framing is rarely used to address maternal health interventions [12]. In this scenario, developing a minimum data set (MDS) can play a crucial role in facilitating standardized care and driving quality improvement [15]. By establishing a consistent framework for data collection, MDSs ensure that critical information is accurately captured and easily accessible. This approach enhances data sharing and communication across healthcare settings and strengthens the reliability and comparability of data reporting in empirical research [16]. However, a consensus on a minimum data set of midwifery interventions using a salutogenic framework is lacking in several countries, including the Italian healthcare context.

This study aims to develop and validate the Midwifery Interventions Classification (MIC), an evidence-based and standardized taxonomy of core midwifery interventions, pursuing a systematic approach to midwifery data collection. The MIC considers a salutogenic perspective in maternity care. The preliminary results of this research project were published in the study protocol [17]. Specifically, the MIC is based on the Donabedian framework, which defines the quality of care through structure, process, and outcomes indicators [17]; the Salutogenic framework, which focuses on a physiological approach to birth, pregnancy, and postnatal care [10]; and the QMNC framework, which outlines the scope of midwifery care [15]. The MIC’s scope includes healthy women with low-risk pregnancies, their newborns, and their families, covering care from the antenatal period through birth and postnatal care.

## 2. Materials and Methods

### 2.1. Design

The study design described the Consensus process up to the validation of the MIC through the Delphi method, consistent with the recommendations of the “COnsensus-based Standards for the selection of health Measurement Instruments” (COMET) Handbook: version 1.0 [18], the Conducting and REporting DElphi Studies (CREDES) guidance [19], and the ACcurate COnsensus Reporting Document (ACCORD) guideline [20]. The Delphi technique is a study design commonly used when evidence on a topic is limited or fragmented. By gathering expert consensus, this method enables the development of clinical practice recommendations [21]. Consequently, consensus-defined criteria and statistical metrics were used to validate the MIC.

The Consensus process was carried out between February 2023 and June 2024, involving three different panels of stakeholders: Midwives, Healthcare Researchers, and Maternity Service Users. The study protocol, including the preliminary results from the Developmental phase, has been previously published [17]. This research project is registered with the Core Outcome Measures in Effectiveness Trials (COMET) Initiative (registration number 1723; available online at https://comet-initiative.org/Studies/Details/1723, (accessed on 1 June 2024)). Figure 1 summarises the overall study design.

### 2.2. Consensus Process

The Consensus process aimed to achieve an agreement on the MIC through a Delphi technique and was based on an established method for reaching consensus among field experts and stakeholders [22]. As shown in Figure 1, the Developmental phase resulted in a preliminary and evidence-based list of 165 midwifery interventions identified from the literature review [17]. More specifically, midwifery interventions are defined as components of maternity care provided by midwives to improve and optimize the health outcomes of women, newborns, and the overall public health of society [14]. To facilitate the consensus process analysis, these midwifery interventions were categorized using a theory-driven approach into three main categories: Direct Midwifery Care (i.e., interventions delivered through direct interaction with the birthing person, the newborn, or the family), Indirect Midwifery Care (i.e., interventions delivered away from the birthing person, newborn, or family but in their interest), and Community Midwifery Care (i.e., interventions delivered to promote and preserve community health).

In the consensus process phase, the Delphi technique was used to gather opinions from stakeholders on the importance of core midwifery interventions through two rounds of consensus. The responses were collected using a web-based survey that was managed and analyzed using SurveyMonkey^®^ software (https://it.surveymonkey.com/). Lastly, a consensus meeting among authors was performed to finalise the consensus on the MIC.

#### 2.2.1. Participants: Panel Composition and Group Size

To integrate different opinions and perspectives, 148 participants from three diverse stakeholder panels were invited to participate in April 2023 through a purposive sampling procedure. The criteria for participant selection were outlined in the previous research protocol [17] and are briefly summarized below. The first panel consisted of Midwives with at least 1 year of experience in midwifery care. The second panel was comprised of healthcare researchers who are experts on midwifery research and the Delphi technique. The third panel included healthy women who experienced a physiological pregnancy and birth as Maternity Service Users within the last 5 years. The participants in each panel were identified through a network of researchers and trained midwives at the University of Pavia (Italy). Participants in the user panel, however, were identified through hospitals affiliated with the University of Pavia. The first author of the study was responsible for reaching out to participants from the contact list. Given the source of these contacts, participants for each panel were primarily from regions in Northern Italy.

Table 1 shows the details of the participants invited, participants who completed the first round, participants who completed both rounds, and the response rate between rounds.

#### 2.2.2. Delphi Rounds and Consensus Meeting 

As outlined in the previous research protocol [17], two Delphi rounds were planned to achieve a consensus among stakeholders. Thus, two rounds of Delphi are required as the minimum standard for the consensus process, and further rounds could be assessed based on the consensus definition reached [23]. Round 1 took place between May and July 2023. In Round 1, participants were asked to assess the importance of each midwifery intervention from the preliminary list with the possibility of adding suggestions or new interventions if deemed appropriate. Participants received via e-mail the link to the electronic questionnaire and a support guide comprising midwifery interventions and their definitions. The subsequent Round 2 took place between October 2023 and February 2024. As the study protocol states, all midwifery interventions have been retained between Round 1 and Round 2 [17]. In Round 2, participants who completed Round 1 received statistical feedback on the previous round results, comprising the average scores for each midwifery intervention aggregated across stakeholder panels. Moreover, each participant received the updated support guide comprising midwifery interventions and their definitions, as well as a personal document containing their scores assigned in Round 1. Then, participants reconsidered their scores and assessed the addition of new interventions. Finally, between May and June 2024, the authors reviewed the emerging list of interventions to finalise their consensus and achieve the final MIC.

#### 2.2.3. Questionnaire and Scoring System

The electronic questionnaire was composed of midwifery interventions alphabetically and divided into three categories: Direct Midwifery Care, Indirect Midwifery Care, and Community Midwifery Care. The 9-point Likert scoring system was used to evaluate each midwifery intervention, as recommended by the Grading of Recommendations, Assessment, Development, and Evaluation (GRADE) Working Group [24]. Specifically, 1 to 3 meant a midwifery intervention was not important, 4 to 6 important but not critical, and 7 to 9 critical. Furthermore, an “unable dot score” category was included for participants who may not have the expertise to rate specific midwifery interventions. Participants could also provide free-text comments in each round to add rationale for their chosen score, suggest modifications to the interventions’ wording, or propose new interventions.

#### 2.2.4. Consensus Definition and Data Analysis

To define consensus, the study protocol had foreseen the “70/15%” method [18] implemented by Wylde et al. (2015) [25]. This method required that over 70% of participants scored each midwifery intervention between 7 and 9, with fewer than 15% scoring it between 1 and 3, in order to retain the intervention across rounds [18]. Moreover, an additional criterion introduced by Wylde et al. was added: midwifery interventions scored as 7–9 by more than 90% of one-panel members were also carried forward to the subsequent round, regardless of the rating of the other panel [25]. The rationale was to include midwifery interventions deemed crucial by most participants and ensure that midwifery interventions considered exceptionally important by only one panel were not omitted. In addition to the consensus definition criteria, based on the percentage agreement rate among the participants, the mean scores for each midwifery intervention were compared among panels and within each panel over time (i.e., across Rounds 1 and 2). Given the number of comparisons “between subjects” and “within subjects”, a mixed ANOVA test was used to assess the variance in responses. All statistics were deemed significant at *p* < 0.05, and the analysis was performed using SPSS version 29.

#### 2.2.5. Validity and Reliability

The validity and reliability of the results derived from this study are linked to the assessment of attrition and attrition bias. Attrition is the degree of non-response after the first Delphi round. As a result, attrition bias occurs when the participants who do not respond in subsequent rounds have different views from their stakeholder peers who continue to participate. The COMET Handbook deems a response rate of around 80% acceptable for each stakeholder panel [18]; therefore, both the overall and response rates for each stakeholder panel have been calculated (Table 1). Moreover, attrition bias was assessed by comparing the frequency distributions of average scores on each point of the Likert scale between those who completed Round 2 and those who dropped out after Round 1. As per the validity and reliability measures, we calculated the Content Validity Index (CVR) and the Intraclass Correlation Coefficient (ICC) for each midwifery intervention included in the second round of Delphi.

### 2.3. Ethical Consideration

All input and involvement during the process remained highly confidential. Based on the information given, each invited participant could either agree to participate in the study or exercise their right to withdraw at any time. Data were managed following the General Data Protection Regulation (GDPR). The institutional review board of the University of Pavia approved the project (n. CD/03/2022).

## 3. Results

### 3.1. Delphi Panel Demographics

As shown in Table 1, of the 148 participants who indicated their willingness to participate in the study, 125 completed Round 1 and were invited to Round 2, and 103 completed both rounds. The overall response rate (82%) aligned with the recommendations, attesting to its value of around 80% [18]. The same had occurred for the Midwives Panel (85%) and the Healthcare Researchers Panel (91%) but did not occur for the Maternity Service Users Panel (71%). Table 2 shows demographics data collected through the Delphi survey for the Midwives Panel, aggregated per Delphi round. For both Delphi rounds, most of the midwives were female, aged 36 years, unmarried, with a Postgraduate education, with a work experience of 5–10 years, and who worked in a Birth centre (public or private). Midwives mainly described the midwifery care offered to the women by the healthcare facility where they work as good.

Table 3 shows demographics data collected through the Delphi survey for the Healthcare Researchers Panel, aggregated per Delphi round. The healthcare researchers were mainly female, aged 41 years, married, with a Postgraduate education, work experience greater than 15 years, and a researcher position.

Table 4 shows demographics data collected through the Delphi survey for the Maternity Service Users Panel, aggregated per Delphi round. Most women were Italian, had a bachelor’s degree, were employed, and had one child. Considering their last pregnancy experience, they mainly stated that they had received excellent midwifery care and support for breastfeeding initiation. Among those who receive midwives’ support to initiate breastfeeding, the majority breastfed for more than 6 months.

### 3.2. Round 1

During Round 1, participants were asked to evaluate the preliminary and evidence-based list of 165 midwifery interventions resulting from the literature review through the Developmental phase (Appendix A) using a 9-point Likert scoring system. As shown in Table 1, Round 1 was completed by 125 participants. Of these, 82 were part of the Midwives Panel, 12 of the Healthcare Researchers Panel, and 31 of the Maternity Service Users Panel (Table 1).

Round 1 achieved an overall high degree of consensus between participants, equal to 81%. Participants reached a consensus on 134 midwifery interventions out of 165. Of these 134 midwifery interventions, 133 were assessed as “critical”, and one (i.e., Cup feeding) was evaluated as “not important”. No consensus was reached on the remaining 31 midwifery interventions. In this regard, significant mean differences between the panels were detected for 85 interventions (see Appendix A). However, the mean scores were interpreted according to the Likert scale used for the consensus definition. Thus, if the mean scores for a midwifery intervention consistently ranged from 7 to 9 between panels, it was considered “critical” by all panels. Conversely, if the mean scores varied across different Likert categories score (i.e., not important, important, critical), variability in responses among panels was assumed. Based on the descriptive statistics in Appendix A, critical mean values were identified for only 20 midwifery interventions, indicating potential intergroup variability.

The authors analysed the suggestions and comments made by Delphi participants in the open-ended questions at the end of the survey (Appendix A). Based on these suggestions and comments, one midwifery intervention was modified in the title (i.e., *Early labour care*) and two midwifery interventions were modified both in titles and definitions (i.e., *Contraception and family planning counselling: Counsel the birthing person and family on contraception and family planning* and *Sexual and reproductive health counselling: Counsel the birthing person on sexual and reproductive health during pregnancy and postnatal period*). Participants also suggested that the lack of two concepts generated two new midwifery interventions. The first midwifery intervention added was *Consideration of the family context* with the proposed definition: *Consider whether the birthing person and/or the family are likely to require support in fulfilling their parental role and signpost to appropriate services* [15]. The second midwifery intervention added was Humanization of care with the proposed definition: *Plan and provide midwifery care based on respect for the dignity, uniqueness, individuality and humanity of the birthing person and newborn. This requires appropriate working conditions, with sufficient human and material resources* [26].

Moreover, four midwifery interventions were modified in their wording by authors to adopt a more inclusive language (i.e., Antenatal education: Promote the birthing person’s ability to activate their skills to cope with labour and childbirth by providing information to support informed choice; Breastfeeding counselling: Counsel the birthing person on breast/chestfeeding, following the 10 steps to successful breastfeeding by WHO/UNICEF; Management of cardiotocographic changes: Implement conservative measures in response to changes in fetal heart rate (FHR) and/or uterine contractions (UC) detected by cardiotocography; Monitoring of term pregnancy: Monitor parental and fetal well-being at the end of pregnancy to plan and deliver appropriate childbirth care). Finally, two midwifery interventions (i.e., Fetal well-being assessment: FHR and Fetal well-being assessment: FM) were merged by the authors for stackable meaning, resulting in a Fetal well-being assessment: Assess fetal well-being through auscultation and/or fetal heart rate (FHR) recording and fetal movement (FM) assessment. At the end of Round 1 analysis, the MIC consisted of 166 midwifery interventions. The overall results from Round 1 and Round 2, and the final decision, are summarised in Appendix A.

### 3.3. Round 2

In Round 2, participants who completed Round 1 were asked to newly assess each of 166 midwifery interventions resulting from Round 1, relying on statistical feedback from Round 1. As described in Table 1, Round 2 was completed by 103 participants with an overall response rate of 82%. After Round 2, 12 participants left the Midwives Panel (response rate 85%), one participant left the Healthcare Researchers Panel (response rate 92%), and nine participants left the Maternity Service Users Panel (response rate 71%). In Appendix A, we have also provided the main demographic characteristics of the non-responders in the second round of the Delphi survey for each panel. Significant differences between the profiles of respondents and non-respondents were found only among midwives and maternal service users. Those who did not respond in Round 2 showed a lower level of quality perception of care delivered in the midwives’ panel (mean 2.79 ± 0.65 vs. 2.25 ± 0.62, *p* = 0.010) and were more likely to be unmarried (*p* = 0.027) in the maternal service user panel.

Round 2 achieved a higher degree of consensus than Round 1, equal to 87%. Participants reached a consensus on 144 midwifery interventions out of 166. All of these 144 midwifery interventions were assessed as “critical”, and among these, two midwifery interventions were added in Round 1. The remaining 22 midwifery interventions that did not reach the consensus for inclusion or exclusion are among those that had not reached the consensus even in Round 1. Therefore, Round 2 results are consistent with the previous results from Round 1. At the end of Round 2 analysis, the MIC consisted of 144 midwifery interventions. Appendix A summarises the overall results. Also in the second Delphi round, mean differences between panels were detected for 30 midwifery interventions (see Appendix A). However, nearly all midwifery interventions showing mean differences across panels were subsequently removed during the final consensus discussion for the MIC version.

We have further assessed if the attrition in Round 2 introduced a bias, conditioning the responses among panels. Thus, we have counted the frequencies of the average scores for each Likert system point, between participants dropping out after Round 1 and those completing Round 2 (Figure 2). The results of those who did not complete Round 2 did not represent extreme views, suggesting that bias was not introduced through attrition between rounds. Furthermore, comparing the mean scores of each midwifery intervention across the two rounds, the lack of significant interactions indicated that the differences in the mean scores between panels remained consistent over time (see statistics in Appendix A).

### 3.4. Validity and Reliability Results

The reliability and validity of the consensus responses were further assessed by calculating ICC and CVR scores for each midwifery intervention in the second Delphi round (n = 166 midwifery interventions). For ICC, high reliability (ICC ≥ 0.80) was achieved among responses within each panel for all midwifery interventions (see Appendix A). Furthermore, Appendix A reports the descriptive statistics for CVR scores for each panel. Given the large sample size, a CVR threshold value of ≥0.30 was considered adequate [27]. Although most midwifery interventions in the second round were deemed essential across all panel groups, critical CVR values were also identified. These results were considered by the authors in the final consensus meeting. Specifically, midwifery interventions with critical CVR scores were those selected for exclusion or modification in the final MIC version.

### 3.5. Consensus Meeting

The resulting MIC from Round 2, comprising 144 midwifery interventions, has undergone a further in-depth review process led by authors with the aim to refine the final MIC. The consensus meeting was held in both Italian and English due to the presence of two authors, native speakers of Italian and English, who were to finalise both versions of the MIC. From the consensus meeting, the authors excluded midwifery interventions that did not reach the consensus after the two rounds (n = 22). Moreover, the authors merged eight midwifery interventions for stackable meaning. Expressly, the midwifery intervention *Learning process facilitation* was annexed to *Promotion of health literacy*, *Guidelines on how to prevent critical situations to Signs and symptoms counselling*, *Role empowerment to Parental role promotion*, *Management of acuity codes to Obstetric triage*, *Environmental well-being management* and *Environmental safety management to Management of the environment*, *Latex use precautions* by *Allergy management* and the respective definitions have been updated (Appendix A). Furthermore, the midwifery intervention *partograph was discussed because international guidelines no longer support* its use [28,29,30]. The concept at the core of this midwifery intervention is to monitor and document maternal-fetal well-being and the progression of labour over time; thus, this midwifery intervention was merged with similar existing interventions of *Maternal-fetal monitoring in labour* and *Clinical record keeping*. Finally, authors excluded one midwifery intervention (i.e., *Venus blood sampling*) for consistency because other interventions referring to specific activities did not reach the consensus and were excluded (e.g., *Intravenous cannulation* or *Massage*).

The final MIC is a validated data set of 135 core midwifery interventions classified into three main categories: Direct Midwifery Care (n = 80 interventions), Indirect Midwifery Care (n = 43 interventions), and Community Midwifery Care (n = 12 interventions) (Appendix A). Direct Midwifery Care refers to midwifery interventions provided through an interaction with women, involving compensatory interventions or supportive actions care to promote women’s physical and mental well-being, and healthy newborn growth. Indirect Midwifery Care, on the other hand, refers to interventions provided away from the woman but in her best interest. Indirect midwifery interventions encompass managerial, organizational, and healthcare professional competence aspects to ensure quality and safe maternal care. Lastly, Community Midwifery Care refers to midwifery interventions that address a broad range of health and social needs for women, families, and the community, considering local health policies and resources. Thus, community midwifery interventions focus on promoting healthy sexual and reproductive health and respectful maternity care at the community level.

## 4. Discussion

This study provided a classification and taxonomy of midwifery interventions according to a salutogenic approach to maternity care. To the best of our knowledge, this study represents the first attempt to provide a comprehensive description and synthesis of salutogenic midwifery care interventions, using evidence-based international guidelines and a consensus process by examining the opinions of a panel of experts on the importance of each midwifery intervention. However, considering the peculiarities of each country regarding the regulation of midwifery practice, professional educational pathways, and models of care, the results of this study have applicability primarily within the Italian context, with the potential for further scaling up at the international level. The final version of the MIC encompasses 135 standardized midwifery interventions, classified into three main categories: Direct Midwifery Care, Indirect Midwifery Care, and Community Midwifery Care.

For direct midwifery interventions (n = 80), the consensus rate among the panel of experts was 83.2%. The midwifery interventions that did not reach consensus (n = 17) primarily concerned alternative feeding methods for newborns (e.g., bottle-feeding, cup feeding) and the use of non-pharmacological methods for pain management during labor (e.g., therapeutic massage and hydrotherapy). Nonetheless, several studies in the literature have highlighted the effectiveness of using non-pharmacological interventions to promote pain management and maternal coping strategies during labor within a framework of salutogenic interventions [31,32]. However, there could be several barriers to delaying or underusing non-pharmacological interventions in labour by midwives, mainly related to health facility factors and resources, health practitioners’ knowledge and competence, and health consumers’ preferences [33]. More specifically, in Italy, where most births take place in hospitals, not all the facilities have the infrastructural capacity or sufficient resources to ensure, for example, waterbirths. Furthermore, no evidence-based guidelines at the national level are available to address and support midwifery practice in delivering non-pharmacological interventions [34].

For indirect midwifery interventions (n = 43), the consensus rate among the panel of experts was equal to 92.3%. This category of interventions emphasizes areas of midwifery practice aimed at ensuring accessible, continuous, personalized, and high-quality care in maternity settings. In other words, these midwifery interventions become crucial for the planning and delivering of clinical care pathways that significantly reduce unnecessary medical interventions in maternity care. The midwifery continuity care framework includes indirect interventions related to a case-management approach, discharge planning, and ensuring integrated and transitional care across services [1]. Accordingly, the recent scoping review by Bradford et al. (2022) provided evidence on the global level of implementation of the midwifery continuity care model [35]. Although midwives play a pivotal role in leading these continuity care models, their implementation was largely prevalent in high-income countries (e.g., Australia and the United Kingdom), while low prevalence was observed in low- and middle-income countries. However, even in high-income countries, scaling up the implementation of the midwifery continuity care model at a national level has not been achieved in any country except New Zealand [35]. Although the midwifery continuity care model is currently underway in the Italian context, the predominant care model remains that of the private gynecologist, chosen by 66% of women [34]. These considerations highlight the organizational, systemic, and professional challenges to sustaining the adoption and feasibility of a care model that recognizes the full scope of midwifery practice in providing direct, indirect, and community midwifery interventions within a continuum of care.

The role of midwives in the public health care context is widely recognized for providing preventive public health interventions during pregnancy and the postnatal period. In fact, for community midwifery interventions (n = 12), the consensus rate among the panel of experts was equal to 92.3%. Community midwifery interventions focus on health education, support, screening, surveillance, and promoting a culture of respectful maternity care [36,37]. These findings align with other studies and policy actions that have investigated the scope of midwifery practice in community care [37,38,39]. In a community health context, these public health interventions primarily aim to address social determinants of health (SDOH) in maternity care to reduce health inequalities and disparities and improve pregnancy and childbirth outcomes [40]. In fact, the literature has highlighted how community-based and continuity midwifery-led care models contribute to preserving a more physiological approach to pregnancy and higher maternal satisfaction compared to hospital care settings, particularly for vulnerable women with high social risk factors and low-risk women in antenatal care [41]. For example, vulnerable women who received a midwifery specialist model of care were more likely to experience skin-to-skin contact and use non-pharmacological approaches to pain relief during labor [41].

The MIC data set for maternity care could be applied across various scopes of implementation, such as in clinical and organizational settings, as well as in advancing research and education. In clinical practice, the use of the MIC will facilitate a standardized data-collection approach for organizing and making health information and practices consistent [15,42]. This standardization aids in clinical documentation, care communication across settings, data integration across systems and services, productivity measurement, and reimbursement of the midwifery interventions. Moreover, standardized classifications provide a foundation for capturing the full scope of care delivered, defining valid measures of workload and staffing levels in organizational contexts [43]. From a research perspective, there is a need for more evidence on the effectiveness of midwifery interventions on salutogenic outcomes, using a taxonomy that enhances comparability of research findings [44]. Additionally, further evidence is required to understand the strategies and determinants for implementing midwifery interventions in real-world care settings. Finally, the MIC can also support education and training programs by providing a precise classification of interventions within conceptual categories [45]. This can guide educators, students, and healthcare workers in delivering salutogenic maternity care.

### Limitations and Strengths

The main limitations of this study are consistent with the employed methodology, as declared in the study protocol. Participants might have perceived the process as time-consuming, increasing the attrition between rounds. For these reasons, given the sufficient overarching consensus reached among the stakeholder panels and the potential risk of attrition bias among respondents, we considered two Delphi rounds sufficient to provide initial evidence of MIC validation. However, further rounds could have improved consensus among stakeholders regarding midwifery interventions. Despite the overall, the Midwives Panel and Healthcare Researchers Panel response rate was consistent with the recommendations, attesting to its value of around 80%; the same did not occur for the Maternity Service Users Panel, attesting to the response rate of around 71%. However, the literature indicates that an overall response rate of 70% can be also considered sufficient to ensure rigor in the Delphi process [46]. The analysis shows that the average score of those who did not complete Round 2 did not represent extreme views, suggesting that bias was not introduced through attrition between rounds. Inevitably, examining average scores between completers and non-completers has its limitations; for example, non-completers may score some midwifery interventions much higher than completers and score other midwifery interventions much lower than completers, but average scores may remain similar between the two groups [18,47]. Other implicit limitations of this approach rely on a restriction of the possibility of elaborating in-depth on the views of the participants as the method does not encompass open discussions. Moreover, since most participants were from regions in Northern Italy, this may limit the ability to achieve a comprehensive representation of views from a national perspective within the panel. Given these potential limitations, we suggest caution in generalizing the results of this study. Finally, reaching a consensus does not necessarily imply achieving the best possible MIC. In this sense, future validation studies (e.g., criterion-related validity studies) are needed to corroborate the MIC, as well as to measure its effectiveness and acceptability in achieving adequate patient-level outcomes. Furthermore, this classification needs to be updated to reflect the evolving role of the midwife and the needs of the users. On the upside, the strength of this study is the adherence to a robust methodology since it is designed on a priori protocol, following and combining the best recommendations for the Delphi method, ensuring reproducibility, transparency, and reliability.

## 5. Conclusions

The MIC is a valid and reliable tool specific to the Italian midwifery care context with the potential for international transferability, implementation, and scale-up. The MIC could boost quality improvement, education, and comparable data collection for research, sustaining midwives’ role in promoting optimal health for women, newborns, and families at large. Thus, using the MIC could enable healthcare providers to support interventions that appropriately address health and social needs in maternity care. Further studies are needed to corroborate the validity of the MIC in relation to measurable patient-level outcomes and the previously developed M-COS.

## Figures and Tables

**Figure 1 healthcare-12-02228-f001:**
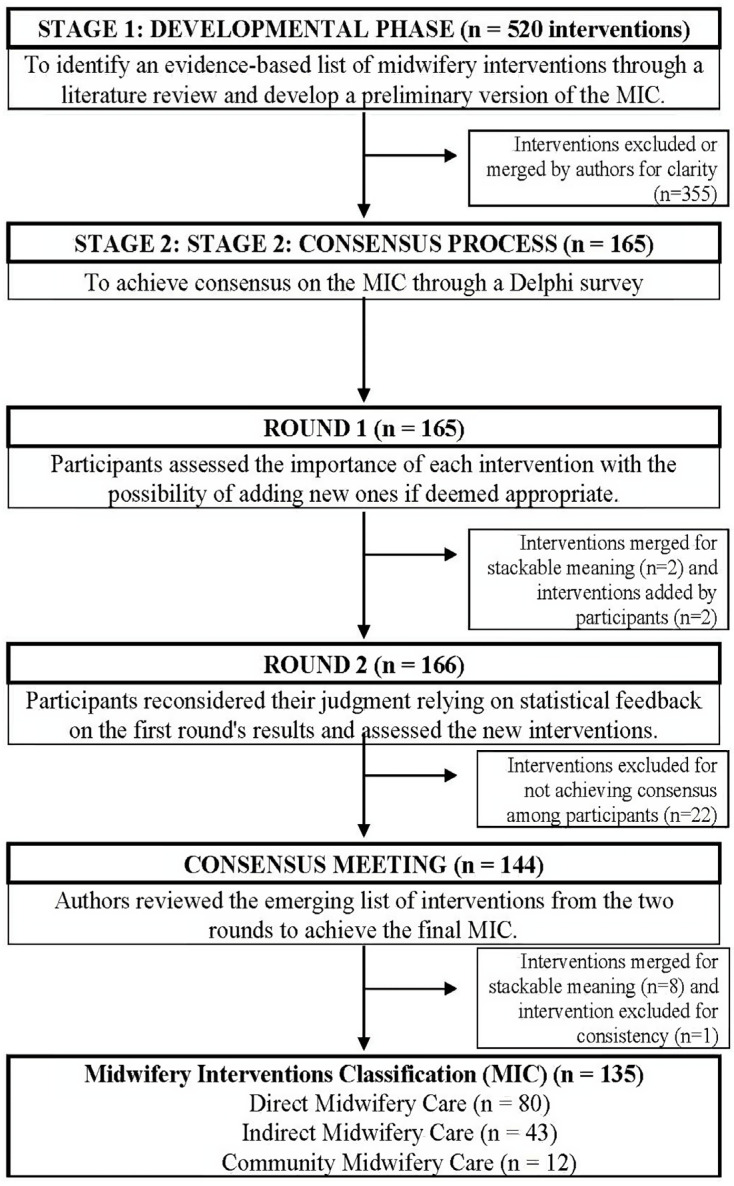
Flow chart of the overall study design.

**Figure 2 healthcare-12-02228-f002:**
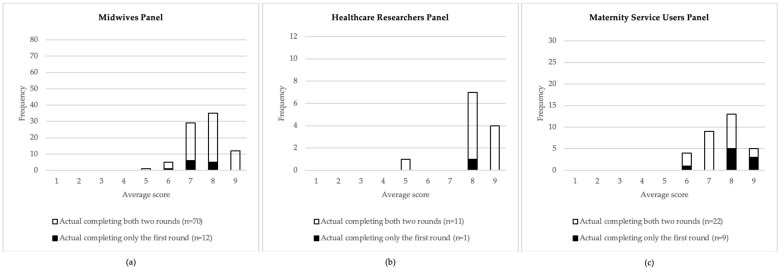
This figure illustrates the distribution of frequencies of the average scores for three distinct panels: (**a**) Midwives Panel, (**b**) Healthcare Researchers Panel, and (**c**) Maternity Service Users Panel. Each panel’s distribution is depicted with bars representing participants who completed both rounds (hollow bars) and those who completed only the first round (solid bars). The x-axis represents the average scores ranging from 1 to 9, while the y-axis indicates the frequency of responses. The figure highlights the variability in scoring across the different panels, with a concentration of higher average scores (7–9) across all panels, particularly in the Healthcare Researchers Panel.

**Table 1 healthcare-12-02228-t001:** Breakdown of participants invited and completing both Delphi rounds.

	Total Invited	Actual Completing the First Round	Actual Completing Both Two Rounds	Response Rate Between Rounds (%)
	Total Participants Invited (n)	Participants Completing the 1° ROUND Delphi (n)	Participants Completing the 2° Round Delphi (n)	Response Rate Between Rounds (%)
Midwives	94	82	70	85%
Healthcare Researchers	12	12	11	92%
Maternity Service Users	42	31	22	71%
Overall Participants	148	125	103	82%

**Table 2 healthcare-12-02228-t002:** Demographics of the Midwives Panel per Delphi rounds.

Characteristics	Round 1 (n = 82)	Round 2 (n = 70)
	May–July 2023	October 2023–February 2024
Gender, n (%)		
Female	80 (98)	69 (99)
Male	2 (2)	1 (1)
Civil status, n (%)		
Married	31 (38)	25 (36)
Not married	51 (62)	45 (64)
Education, n (%)		
Old High School Leaving Qualification in Nursing and Midwifery *	5 (6)	4 (6)
Bachelor’s Degree	33 (40)	23 (33)
Postgraduate Education	44 (54)	43 (61)
Work experience, n (%)		
1–5 years	21 (26)	17 (24)
5–10 years	23 (28)	22 (31)
10–15 years	16 (20)	15 (21)
>15 years	22 (27)	16 (23)
Workplace, n (%)		
Birth centre (public or private)	65 (79)	55 (79)
1st level **	22 (34)	18 (33)
2nd level **	44 (68)	37 (67)
Health district (public or private)	14 (17)	13 (19)
Other ***	3 (4)	2 (3)
*Centre for Medically Assisted Procreation*	*1 (33)*	*1 (50)*
*University*	*2 (67)*	*1 (50)*
If you work in a healthcare facility, how would you describe the midwifery care offered to the women? n (%)		
Poor	4 (5)	3 (4)
Appropriate	22 (28)	17 (25)
Good	50 (63)	45 (65)
Excellent	6 (8)	4 (6)
Age (years) (M ± SD)	36 ± 9	36 ± 9

* To date equivalent to a bachelor’s degree. ** In accordance with “Accordo Stato-Regioni 16 December 2010—Allegato 1b”, 1st level birth centre: 500–1000 births per year ≥ 34 weeks; 2nd level birth centre: >1000 births per year whether maternal and fetal risk levels. *** Other workplace categories are marked in italics.

**Table 3 healthcare-12-02228-t003:** Demographics of the Healthcare Researchers Panel per Delphi rounds.

Characteristics	Round 1 (n = 12)	Round 2 (n = 11)
	May–July 2023	October 2023–February 2024
Gender, n (%)		
Female	10 (83)	9 (82)
Male	2 (17)	2 (18)
Civil status, n (%)		
Married	6 (50)	6 (55)
Not married	6 (50)	>5 (45)
Education, n (%)		
High School Graduation	0 (0)	0 (0)
Bachelor’s Degree	0 (0)	0 (0)
Postgraduate Education	12 (100)	>11 (100)
Work experience, n (%)		
1–5 years	1 (8)	1 (9)
5–10 years	2 (17)	2 (18)
10–15 years	3 (25)	3 (27)
>15 years	6 (50)	>5 (45)
Job position, n (%)		
University Professor	1 (8)	1 (9)
Researcher	7 (58)	7 (64)
Research Fellow	0 (0)	0 (0)
Other *	4 (33)	3 (27)
*Degree Course Director*	*2 (50)*	*2 (67)*
*Staff Midwife*	*1 (25)*	*1 (33)*
*Senior Lecturer Academic*	*1 (25)*	*0 (0)*
Age (years) (M ± SD)	41 ± 13	41 ± 14

* Other workplace categories are marked in italics.

**Table 4 healthcare-12-02228-t004:** Demographics of the Maternity Service Users Panel per Delphi rounds.

Characteristic	Round 1 (n = 31)	Round 2 (n = 22)
	May–July 2023	October 2023–February 2024
Gender, n (%)		
Female	31 (100)	22 (100)
Nationality, n (%)		
Italian	30 (97)	22 (100)
Not Italian	1 (3)	0 (0)
Civil status, n (%)		
Married	23 (74)	19 (86)
Not married	8 (26)	3 (14)
Education, n (%)		
Middle School Graduation	0 (0)	0 (0)
High School Graduation	11 (35)	7 (32)
Bachelor’s Degree	20 (65)	15 (68)
Occupation, n (%)		
Employed	26 (84)	18 (82)
Not employed	5 (16)	4 (18)
Number of children, n (%)		
One child	19 (61)	12 (55)
Two children	6 (19)	6 (27)
More than two children	5 (16)	4 (18)
Considering the last pregnancy experience, how would you describe the quality of the received midwifery care? n (%)		
Poor	1 (3)	1 (5)
Appropriate	1 (3)	0 (0)
Good	12 (39)	9 (41)
Excellent	17 (55)	12 (55)
Considering the last pregnancy experience, did you receive midwives’ support for breastfeeding initiation? n (%)		
Yes	20 (65)	13 (59)
No	9 (29)	8 (36)
If you say yes, how much time did you breastfeed? n (%)		
<6 months	8 (40)	4 (31)
6–12 months	6 (30)	4 (31)
Other **	6 (30)	5 (38)
*>12 months*	*3 (50)*	*2 (40)*
*Still breastfeeding*	*3 (50)*	*3 (60)*
Age (years) (M ± SD)	35 ± 4	36 ± 4

Data missing in Round 1: Considering the last pregnancy experience, did you receive midwives’ support for breastfeeding initiation? (n = 2); Number of children (n = 1); Data missing in Round 2: Considering the last pregnancy experience, did you receive midwives’ support for breastfeeding initiation? (n = 1). ** Other breastfeeding categories are marked in italics.

## Data Availability

Data will be available from the corresponding author upon reasonable request.

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
