# Peer review of "Development and Validation of the Midwifery Interventions Classification for a Salutogenic Approach to Maternity Care: A Delphi Study"

_healthcare, 2024, doi:10.3390/healthcare12222228_

Round 1

Reviewer 1 Report

Comments and Suggestions for Authors

Dear Authors

Thanks for your practical study. There are some points better to be considered;

1-What were your criteria for choosing an expert for consensus?

2-Since obstetricians have an important role in maternity care especially in complicated labor why didn't you include this group in participants?

3-How did you prevent bias in experts’ judgments?

4-By means of your consensus what was added to maternity care? please clarify it in conclusion.

Author Response

Comments 1:-What were your criteria for choosing an expert for consensus?

Response 1: Thank you for your comment. The criteria were specified in our previously published study protocol [1]. We have also briefly outlined these criteria in the current manuscript at line 130 on page 4. The criteria are as follows: a) The healthcare professionals panel includes midwives with at least one year of experience in midwifery care. b) The researchers panel includes academic professors or researchers specializing in midwifery or nursing science, with expertise in research methodologies. c) The maternity services users panel includes healthy women who experienced a physiological pregnancy and birth within the past five years.

References:

1.     Maga, G.; Arrigoni, C.; Brigante, L.; Cappadona, R.; Caruso, R.; Daniele, M.A.S.; Del Bo, E.; Ogliari, C.; Magon, A. Developmental Strategy and Validation of the Midwifery Interventions Classification (MIC): A Delphi Study Protocol and Results from the Developmental Phase. Healthcare 2023, 11, 919. https://doi.org/10.3390/healthcare11060919

Comments 2: Since obstetricians have an important role in maternity care especially in complicated labor why didn't you include this group in participants?

Response 2: Thank you for your comment. This is an important point to clarify. Obstetricians, like other healthcare professionals, play a crucial role in providing a multidisciplinary approach to maternity care. However, our decision to involve primarily midwives aligns with our aim to develop and validate a classification of midwifery-led interventions from a salutogenic perspective. This approach focuses on excluding interventions related to the management of complications or pathological conditions, where the role of obstetricians becomes essential. Our aim, therefore, was to highlight interventions that allow midwives to fully exercise their scope of practice. This study represents a first attempt to develop and validate a classification and taxonomy of interventions that reflect the full scope of midwives’ practice in promoting a salutogenic approach to maternity care. However, future research could expand this classification to include an integrated and multidisciplinary perspective that accounts for pathological or high-risk conditions in maternity care.

Comments 3: How did you prevent bias in experts’ judgments?

Response 3: Thank you for your comment. In response to reviewer 3, we adopted several approaches and measures to ensure rigor and validity in the consensus process. Therefore, attrition bias does not appear to be influenced by dropouts across rounds, particularly within the service users’ panel. Furthermore, it is important to highlight an inherent strength of the Delphi study design: this approach involves independent and anonymous assessments by participants, which helps to prevent any potential bias that could influence the panel’s judgments.

Comments 4: By means of your consensus what was added to maternity care? please clarify it in conclusion.

Response 4: Thank you for your comment. We have edited the text in the conclusion paragraph.

Reviewer 2 Report

Comments and Suggestions for Authors

Summary:

This is a Delphi consensus study to develop a classification system for midwifery care in Italy. This adds to the literature in defining the roles of midwifery care which can be used both in future research and clinical care and involved multiple stakeholders in the process which is a strength of the study. 

General comments:

Overall, this manuscript is well written using a well defined method for achieving consensus, if not a bit wordy. It is unfortunate that the final list is so long that it had to be put in a supplementary table instead of in the main text. 

Specific comments:

- In the introduction, I recommend early on explaining what the "salutogenic approach to maternity care" means early on for the reader who may not be familiar with this word and its meaning. Specifically, I would suggest on pay 2, line 51 adding in a phrase such as "focusing on factors supporting human health and well-being rather than factors causing disease" just after "promoting a salutogenic approach to maternity care". 

- Recommend adding the word "such" to line 47 on page 2

Recommend rewording line 54 on page 2 to "majority of maternity care quality indicators refer to prevention of disease or adverse events" for clarity

- Please add where the identification of individuals for the purposive sampling came from - did you search a website of registered midwives in Italy or tried to pick individuals from different regions in Italy? How about the research people - where did you find that list of names? How did you find the maternity services users - through specific clinics or social media or other and please demonstrate whether these people came from different regions in Italy or only urban or rural areas etc. This is what is missing from the methods and tables 2-4

- Table 1. Columns need to be labelled. I do not understand what these numbers represent without reading the text and a table should be interpretable on its own

- Page 5 line 166 this sentence needs rewording as it does not make sense. Is there a comma in the incorrect place or something missing?

- Tables 2 and 3 specifically include "nonbinary" and "prefer not to say" even though none of the participants identified in either of these categories, but table 4 has gender missing. If these are important variables to include then please add to Table 4 as there is a lack of consistency and one assumes these categories were included because there was at least one person who identified with one of these categories.  

-Figure 1. Consider adding the 3 categories (Direct, indirect and community midwifery care) to this flow diagram with the number of items in each category

- General comments on the discussion: the discussion is a bit lengthy and could be cut down on pages 11-12 where it is somewhat repetitive in nature

Author Response

Comments 1: In the introduction, I recommend early on explaining what the "salutogenic approach to maternity care" means early on for the reader who may not be familiar with this word and its meaning. Specifically, I would suggest on pay 2, line 51 adding in a phrase such as "focusing on factors supporting human health and well-being rather than factors causing disease" just after "promoting a salutogenic approach to maternity care

Response 1: Thank you for your comment. We have edited the text as request.

Comments 2: Recommend adding the word "such" to line 47 on page 2

Response 2: Thank you for your comment. We have edited the text as request.

Comments 3: Recommend rewording line 54 on page 2 to "majority of maternity care quality indicators refer to prevention of disease or adverse events" for clarity

Response 3: Thank you for your comment. We have edited the text as request.

Comments 4: Please add where the identification of individuals for the purposive sampling came from - did you search a website of registered midwives in Italy or tried to pick individuals from different regions in Italy? How about the research people - where did you find that list of names? How did you find the maternity services users - through specific clinics or social media or other and please demonstrate whether these people came from different regions in Italy or only urban or rural areas etc. This is what is missing from the methods and tables 2-4

Response 4: Thank you for your comment. We have revised the text as requested on row 135, page 4. Specifically, the participants in the midwives and researchers panels were identified through a network associated with the University of Pavia (Italy), while participants in the user panel were identified through hospitals affiliated with the University of Pavia. The first author of the study was responsible for reaching out to participants from the contact list. Given the origin of these contacts, participants in each panel were primarily from regions in Northern Italy. However, we lack sufficient information to specify participants' specific regions. Consequently, we have noted this as a potential limitation of our study on row 472, page 14, as it may limit the ability to capture a national perspective.

Comment 5:. Table 1. Columns need to be labelled. I do not understand what these numbers represent without reading the text and a table should be interpretable on its own

Comments 6: Page 5 line 166 this sentence needs rewording as it does not make sense. Is there a comma in the incorrect place or something missing?

Response 6: Thank you for your comment. We have edited the text as request.

Comments 7:  Tables 2 and 3 specifically include "nonbinary" and "prefer not to say" even though none of the participants identified in either of these categories, but table 4 has gender missing. If these are important variables to include then please add to Table 4 as there is a lack of consistency and one assumes these categories were included because there was at least one person who identified with one of these categories.  

Response 7: Thank you for your comment. We have edited Tables 2 and 3 accordingly. We apologize for the oversight, as the inclusion of the sub-categories’ 'nonbinary' and 'prefer not to say' under the gender variable was an error. Consequently, we do not have data for these categories and can only provide a gender breakdown for the two sub-categories: male and female. For the panel of maternity service users, since it focuses on including women experiencing physiological pregnancy and childbirth, the criterion limits the assessment to female participants, thereby excluding other family members and individuals of different genders.

Comments 8: Figure 1. Consider adding the 3 categories (Direct, indirect and community midwifery care) to this flow diagram with the number of items in each category

Response 8: Thank you for your comment. We have edited Figure 1 as request.

Comments 9: General comments on the discussion: the discussion is a bit lengthy and could be cut down on pages 11-12 where it is somewhat repetitive in nature

Response 9: Thank you for your comment. We have edited the text as request.

Reviewer 3 Report

Comments and Suggestions for Authors

Line 44: The authors' article contains sections that do not adhere to the journal's citation format (e.g.; Sandall et al. (2024) established...).

Line 135-161: While the use of the Delphi method for developing the Midwifery Interventions Classification (MIC) is logical, there are several limitations in its application here. First, the study's reliance on a two-round Delphi process may not allow for the depth of exploration required for such a nuanced topic. A more iterative and longer process could yield more robust results. Additionally, the response rate for the Maternity Service Users panel, at 71%, is lower than the recommended 80%, raising concerns about potential bias. Although the study suggests that this did not affect the results, a deeper analysis of non-respondents' views is necessary. Lastly, the design does not incorporate any open-ended discussions, which limits the opportunity for participants to fully elaborate on their perspectives.

Line 216: There are discrepancies in the total number of panelists reported in Table 4 for Round 1. For example, the number of children is listed as "One child: 19, Two children: 6, More than two children: 5," which totals 30. This suggests that one panelist either has no children or was excluded from the data. Please correct this inconsistency.

 Line 279-285: The authors presented the distribution of Likert 9-point scale scores for the three panel groups in Figure 2. However, to compare the differences between the groups, an ANOVA should be conducted to test for statistical significance. Additionally, post-hoc tests are necessary to determine whether intergroup differences are statistically significant, and p-values should be reported to indicate the significance of these statistics. Without following these procedures, merely showing differences in skewness through an intuitive graphical comparison is insufficient for proper result analysis. This omission undermines the rigor of the statistical comparison.

Line 307-320: The absence of Content Validity Ratio (CVR) values in the authors' Delphi study raises concerns about the transparency and rigor of the validation process. The CVR is a widely recognized statistical measure used to assess the degree of expert agreement on the relevance of specific items in a study. By omitting CVR values, the authors fail to provide a clear quantitative measure of content validity, which could undermine the reliability of the consensus reached. Including CVR values would have strengthened the study’s methodological rigor by demonstrating a higher level of objective expert agreement.

Line 410-412: The authors only employed two Delphi rounds, which may not be sufficient to achieve true consensus on a complex topic like midwifery interventions. For a topic involving multiple stakeholders, more iterative rounds are often necessary to refine ideas and ensure that a robust consensus is reached.

Author Response

Comments 1: Line 44: The authors' article contains sections that do not adhere to the journal's citation format (e.g.; Sandall et al. (2024) established...).

Response 1: Thank you for your comment. We have updated the references in accordance with the journal's guidelines. The changes in the text are highlighted in red.

Comments 2: Line 135-161: While the use of the Delphi method for developing the Midwifery Interventions Classification (MIC) is logical, there are several limitations in its application here. First, the study's reliance on a two-round Delphi process may not allow for the depth of exploration required for such a nuanced topic. A more iterative and longer process could yield more robust results. Additionally, the response rate for the Maternity Service Users panel, at 71%, is lower than the recommended 80%, raising concerns about potential bias. Although the study suggests that this did not affect the results, a deeper analysis of non-respondents' views is necessary. Lastly, the design does not incorporate any open-ended discussions, which limits the opportunity for participants to fully elaborate on their perspectives.

Response 2:  Thank you for your comment. We agree that additional rounds of the Delphi process could likely have further improved consensus among stakeholders on the topics ‘intervention. However, several methodological reasons led us to consider two Delphi rounds adequate, as briefly explained below. Firstly, consistent with the guidelines in the Delphi method user manual, we adhered to the minimum required rounds, which is two (Fink-Hafner et al., 2019; Niederberger et al., 2021). Thus, two round Delphi can be considered sufficient if a preliminary literature review has been conducted to guide the consensus process (see the previous publication of the study protocol) (Maga et al., 2023). Furthermore, the need to conduct additional rounds depends on the level of consensus reached within these two rounds, as the information obtained initially directly influences the subsequent stages of the Delphi process. In our case, the authors agreed that the consensus reached among stakeholders after two rounds was sufficient, reaching a satisfactory consensus. It is also important to consider the challenges in conducting additional rounds Delphi. Among these, participant engagement over the rounds presents a notable challenge, especially in voluntary research studies, as in our case. Moreover, detecting a low response rate over the first two rounds is suggested as a criterion to be cautious when considering additional rounds, due to the risk of increasing attrition bias and potentially undermining the validity of the results (Fink-Hafner et al., 2019; Niederberger et al., 2021). For all these reasons, we considered a two-round Delphi approach sufficient and feasible to meet the primary aim of our study. We have added and specify these criteria in the methodological section (at row 145 on page 4) and in the limitation paragraph (at row 455 on page 14) of the manuscript.

Regarding the response rate across Delphi rounds, several recommendations are available about acceptable percentage rates. While we mainly adhered to the user manual's recommendations for designing and conducting the methodological study (Niederberger et al., 2021), various studies in the literature provide updated evidence on acceptable response rate ranges that we can consider. Thus, a response rate of 70% can also be considered sufficient to maintain the rigor of the Delphi technique (Akins et al., 2005). At descriptive level, we have added supplementary file 5 (Table S5) about the socio-demographic characteristics of the non-responders in the second Delphi round by panel.

Finally, we agree that an open-ended discussion with panelists could add value to the study results; however, this was not foreseen in the research protocol, and this limitation has been declared in the manuscript. Nonetheless, in the survey questionnaire, each participant had the opportunity to add free-text comments, which were considered in the revision across the rounds. The responds are reported in the Supplementary 3. Considering the methodological challenges highlighted by the reviewer, which are common and inherent in the Delphi process study design, we have also specified in the limitation paragraph of the manuscript that these methodological issues warrant caution in generalize the results of this study.

Comments 3: Line 216: There are discrepancies in the total number of panelists reported in Table 4 for Round 1. For example, the number of children is listed as "One child: 19, Two children: 6, More than two children: 5," which totals 30. This suggests that one panelist either has no children or was excluded from the data. Please correct this inconsistency.

Response 3: Thank you for your comment. We confirm that there were missing data. We have specified this in the legend of Table 4.

Comments 4: Line 279-285: The authors presented the distribution of Likert 9-point scale scores for the three panel groups in Figure 2. However, to compare the differences between the groups, an ANOVA should be conducted to test for statistical significance. Additionally, post-hoc tests are necessary to determine whether intergroup differences are statistically significant, and p-values should be reported to indicate the significance of these statistics. Without following these procedures, merely showing differences in skewness through an intuitive graphical comparison is insufficient for proper result analysis. This omission undermines the rigor of the statistical comparison.

Response 4: Thank you for your comment. Considering the analytical aim refers to Figure 2, we want to provide more specific information for a correct interpretation. To provide evidence for the absence of attrition bias due to responder drop-out in each group, we counted the frequencies of the mean scores for each possible Likert score. For example, for a Likert score of 1, we counted how many mean values are present among the responses of each group. Thus, we did not make a comparison between the means across groups; instead, we counted the frequencies of the means within each group for each Likert score, between the first and second rounds. To make this process clearer, we changed the wording in the legend of Figure 2. However, we agree that providing only the frequency counts of the means may not be sufficient to justify the absence of the attrition bias and could undermine the process validity.

Thus, we conducted a mixed ANOVA to compare mean scores both between and within groups over time (i.e., Round 1 and Round 2), as described in the methodology section. In Supplementary File 2, we provide a table with p-values for three main factors: category, time, and the category*time interaction. Descriptive statistics and the syntax used for reproducibility are also included.

The "category" factor refers to mean differences between groups. We identified significant differences between groups (p < .05) for 85 items, indicating variability in responses among groups. However, these mean differences among groups should be interpreted using the consensus criteria set by the Likert score system that we referred. Specifically, when mean scores range from a Likert score of 7 to 9, even if intergroup differences, the item may still be considered "critical" by all groups. Conversely, if the mean scores range across different Likert scores (i., e., not important, important, critical), a variability in the responses among groups should be considered. Based on this criterion, 32 items on 85 items initially identified were potentially identified as critical due to significant intergroup differences, across both rounds. In Supplementary File 2, we have also provided descriptive statistics, highlighting in bold the items with a significant mean difference. Notably, almost all items showing significant mean differences across groups were subsequently excluded or modified in the final MIC version.

The "time" factor reflects changes in mean scores over time (between Round 1 and Round 2), independently of the group. Therefore, for most items, the differences in mean scores across time were not statistically significant, suggesting that the scores remained stable between the two rounds. Finally, the "interaction" factor (category*time) assesses whether the change in scores over time differs among groups. In this last case, the lack of significant interactions indicates that mean scores differences among groups did not significantly vary over time, reinforcing that the differences between the two rounds are generally consistent across groups.

Comment 5: Line 307-320: The absence of Content Validity Ratio (CVR) values in the authors' Delphi study raises concerns about the transparency and rigor of the validation process. The CVR is a widely recognized statistical measure used to assess the degree of expert agreement on the relevance of specific items in a study. By omitting CVR values, the authors fail to provide a clear quantitative measure of content validity, which could undermine the reliability of the consensus reached. Including CVR values would have strengthened the study’s methodological rigor by demonstrating a higher level of objective expert agreement.

Comments 6: Line 410-412: The authors only employed two Delphi rounds, which may not be sufficient to achieve true consensus on a complex topic like midwifery interventions. For a topic involving multiple stakeholders, more iterative rounds are often necessary to refine ideas and ensure that a robust consensus is reached.

Response 6: Thank you for your comment. Please refer to the response provided on the comment one.

Brief references list for each comment:

Akins, R. B., Tolson, H., & Cole, B. R. (2005). Stability of response characteristics of a Delphi panel: Application of bootstrap data expansion. BMC Medical Research Methodology, 5(1), 37. https://doi.org/10.1186/1471-2288-5-37

Ayre, C., & Scally, A. J. (2014). Critical Values for Lawshe’s Content Validity Ratio: Revisiting the Original Methods of Calculation. Measurement and Evaluation in Counseling and Development, 47(1), 79–86. https://doi.org/10.1177/0748175613513808

Diamond IR, Grant RC, Feldman BM, Pencharz PB, Ling SC, Moore AM, & Wales PW. (2014). Defining consensus: A systematic review recommends methodologic criteria for reporting of Delphi studies. J Clin Epidemiol, 67(4). https://doi.org/10.1016/j.jclinepi.2013.12.002

Fink-Hafner, D., Dagen, T., Doušak, M., Novak, M., & Hafner-Fink, M. (2019). Delphi method: Strengths and weaknesses. Advances in Methodology and Statistics, 16(2). https://doi.org/10.51936/fcfm6982

Maga, G., Arrigoni, C., Brigante, L., Cappadona, R., Caruso, R., Daniele, M. A. S., Del Bo, E., Ogliari, C., & Magon, A. (2023). Developmental Strategy and Validation of the Midwifery Interventions Classification (MIC): A Delphi Study Protocol and Results from the Developmental Phase. Healthcare, 11(6), 919. https://doi.org/10.3390/healthcare11060919

Niederberger, M., Köberich, S., & members of the DeWiss Network. (2021). Coming to consensus: The Delphi technique. European Journal of Cardiovascular Nursing, 20(7), 692–695. https://doi.org/10.1093/eurjcn/zvab059

Round 2

Reviewer 3 Report

Comments and Suggestions for Authors

#1. The authors utilized the Delphi technique to achieve "expert" consensus on midwifery interventions. However, further discussion should be provided on the rationale for using the Delphi survey. Specifically, the authors could elaborate on why this method was chosen over other consensus-building techniques and discuss its limitations and strengths in relation to the study’s objectives.

#2. Despite measures to mitigate attrition bias, the authors acknowledged that non-completers in Round 2 could have held different views. A more in-depth analysis of reasons for dropout and impact on results would enhance confidence in the findings.

#3. The authors reported a relatively lower response rate for the Maternity Service Users Panel (71%) compared to other groups. This discrepancy could result in an underrepresentation of patient perspectives, which are crucial in assessing the relevance and acceptability of midwifery interventions.

#4. The authors' conclusions about the MIC’s validity are based on consensus, but criterion-related validity studies are needed to determine how well the MIC aligns with patient-level outcomes, which are critical in measuring its practical effectiveness.

Author Response

Comments 1: The authors utilized the Delphi technique to achieve "expert" consensus on midwifery interventions. However, further discussion should be provided on the rationale for using the Delphi survey. Specifically, the authors could elaborate on why this method was chosen over other consensus-building techniques and discuss its limitations and strengths in relation to the study’s objectives.

Response 1: Thank you for your comment, which gives us the opportunity to clarify our choice to use the Delphi technique. While we recognize the methodological limitations of the Delphi technique, it is widely used in the literature and is also recommended by the COMET guidelines for developing a core outcome set (COS) and addressing content validity (1). Moreover, when evidence on a specific topic is limited or fragmented, the Delphi technique plays a crucial role in gathering expert consensus and providing recommendations for clinical practice (2). We have specified it at row 98 on page 3. Specifically for the Italian national context, this study is the first attempt to gather evidence on midwifery interventions from a salutogenic perspective on maternity care, based on expert consensus. Consequently, the results of this study offer a foundation for future empirical studies to further validate the effectiveness of these interventions on specific outcomes.

1. Williamson, P.R.; Altman, D.G.; Bagley, H.; Barnes, K.L.; Blazeby, J.M.; Brookes, S.T.; Clarke, M.; Gargon, E.; Gorst, S.; Harman, N.; et al. The COMET Handbook: Version 1.0. Trials 2017, 18, 280, doi:10.1186/s13063-017-1978-4.

2. Barrett, D.; Heale, R. What Are Delphi Studies? Evid Based Nurs 2020, 23, 68–69, doi:10.1136/ebnurs-2020-103303.

Comments 2: Despite measures to mitigate attrition bias, the authors acknowledged that non-completers in Round 2 could have held different views. A more in-depth analysis of reasons for dropout and impact on results would enhance confidence in the findings.

Response 2: Thank you for your comment. In exploring potential reasons for dropout among non-respondents in the second round of the Delphi study, we used inferential statistics to assess significant differences based on socio-demographic characteristics and descriptive variables for each group. For the researcher group, no significant differences were observed. However, within the users and midwives panels, there were some differences between respondents and non-respondents in the second round. In the midwives group, a statistically significant difference was found in the perception of care quality among respondents in their routine clinical setting. Specifically, those who did not respond in Round 2 had a lower level of quality perception (2.79 ± 0.65 vs. 2.25 ± 0.62, p-value = 0.010). Meanwhile, for the users of maternity services, a statistically significant difference was identified only for marital status variable; non-respondents in Round 2 were more likely to be unmarried (p-value = 0.027). We have edited the text at row 298 on page 10. These two differences, in the midwives and users panels respectively, may not be sufficient to explain the real reasons for dropout or to fully account for participants’ responses. Therefore, the authors agree that the characteristics of non-responders are likely to have a minimal impact on the consensus results.

Furthermore, by adopting a triangulation approach in the previous review round—adding validity statistics such as CVR and ICC to the consensus definition criteria—we are confident that the consensus reached among experts for each item is both reliable and valid.

Comments 3: The authors reported a relatively lower response rate for the Maternity Service Users Panel (71%) compared to other groups. This discrepancy could result in an underrepresentation of patient perspectives, which are crucial in assessing the relevance and acceptability of midwifery interventions.

Response 3: Thank you for your comment. We agree with you. Future studies could therefore use a qualitative approach to assess the acceptability and relevance of these interventions from the patients' perspective or from any other potential users, such as educators. We have specified this issue in the limitations section, stating that the results of this study may not be generalizable.

Comment 4: The authors' conclusions about the MIC’s validity are based on consensus, but criterion-related validity studies are needed to determine how well the MIC aligns with patient-level outcomes, which are critical in measuring its practical effectiveness.

Response 4: Thank you for your comment. We agree that additional psychometric properties of the MIC's validity, such as criterion validity, should be assessed. We have outlined these future directions in the manuscript.
